# Innovative Polycomposite Fertilizer Obtained by Recycling and Processing Three Organic Wastes

**Anca Rovena Lacatusu** [1] (ID)**, Aurelia Meghea** [2]**, Adina Elena Rogozea** [2] **and Carolina Constantin** [2,*]

[1] National R&D Institute for Soil Science, Agrochemistry and Environment-ICPA-Bucharest, 011464 Bucharest, Romania; anca.lacatusu@gmail.com

[2] Faculty of Applied Chemistry and Materials Science, Department of Inorganic Chemistry, Physical Chemistry and Electrochemistry, University Politehnica of Bucharest, 1–7 Polizu, Sector 1, 011061 Bucharest, Romania; a.meghea@gmail.com (A.M.); adinarogozea@gmail.com (A.E.R.)

[*] Correspondence: caraconstantin1964@gmail.com

**Abstract:** The paper aims at testing an innovative organic fertilizer obtained from waste by processing a mixture of marine algae biomass, sewage municipal sludge and farmyard manure. Design of this polycomposite fertilizer is based on adequate conceptual and experimental models by taking into account the complex interactions among these three biomasses. In the first step a detail physico-chemical analysis has been performed on the composition of the three raw materials and also on the soil. In the second phase similar analyses have been carried out on representative samples of soil treated with the compost as compared with untreated soil samples. Analytical methods applied were FT-IR spectroscopy in correlation with organic/inorganic and total carbon (TOC/TIC/TC) analysis. The efficiency of applying this compost on the field at large scale has been assessed by means of fatty acid content of the oleaginous plants cultivated. Based on correlation between production quality and chemical composition of the composted soil, the optimal proportions of the mixture of the three organic wastes will be selected for designing an eco-friendly fertilizer able to improve agrochemical properties of the soil.

**Keywords:** biocomposite fertilizer; sewage sludge; marine algae; farmyard manure; compost; field experiments





## 1. Introduction

The soil degradation process, such as: erosion, decrease of soil organic matter, soil contamination (for example, with heavy metals, pesticides, chemical fertilizer), waterproofing, soil compaction, decrease of the soil biodiversity, soil aggregate breakdown and soil salinization can cause the loss of the soil capacity to fulfill its principal functions. This kind of degradation process can lead to the application of some inadequate agricultural practices, such as: unbalanced fertilization, impoverishment of the underground waters, inappropriate usage of pesticides, utilization of heavy machines, overgrazing, or abandonment of some agricultural practices, as for example, traditional systems of crop rotation practices that have permanently contributed to the restoration of the soil organic matter content. To all of these, may be added desertification, a phenomenon caused by climate change.

Agri-environmental measures offer opportunities to encourage the recovery of soil organic matter, increasing biodiversity, reducing soil erosion as well as contamination and compaction. These measures include support for organic farming, conservative tillage, protection and maintenance of terraces, safer pesticide use, integrated crop management, management of low-intensity pasture, livestock density reduction and the use of certified compost [1]. With the Common Agriculture Policy (CAP) reform of the European Union from 2003, the compliance in order to receive subsidies include respecting the standards of good agricultural and environmental practices relating to the protection of soil against erosion, organic matter preservation and maintenance of soil structure.

Bioproductivity of agricultural cultures was improved by different natural organic and inorganic amendments such as natural zeolites and brown coals [2], gypsum and carbon [3], organic compost produced from slaughterhouse waste [4], tillage and ash, volcanic rock waste [5], biochar and straw [6], biosolids [7], microalgal biomass [8], farmyard manure and crop residues [9], pig manure [10], etc.

The decline of the soil fertility and productivity has stimulated interest in the general improvement of the quality of the soil by adding organic amendments from different sources. To maintain or to improve the fertility and productivity of the agricultural soils, many types of waste can be used, such as: organic municipal waste, sewage sludge, waste from agricultural crops, manure from animals and some types of industrial wastes as a source of organic matter [11]. The use of the fermented manure is a traditional practice in the Romanian agriculture, its importance consists not only in the quantity of nutrients supplied to the soil, but also in improving the growth and development of plants, increasing the capacity of the soil to retain the nutrients, preventing the leaching through the depth of profile with the danger of reaching ground waters, and making them available to the plants according to their consumer needs. The application of the manure on soil helps in improving its physical properties, through the intake of humic substances and microorganisms. The alkaline substances from the manure composition reduce the soil acidity with 0.5 up to 0.8 units of pH when using quantities of 20–30 tones/ha, the effect of fertility soil amelioration is felt far from the fourth year of manure application, while creating the best conditions for increasing efficiency of mineral fertilizers. However, both the livestock reduction in the past 20 years, as well as the insufficient quantity of nutrients in this waste have led to the necessity to use other sources of organic waste, as the sludge from wastewater treatment plants [12].

The focus of investigations into the agricultural value of sewage sludge has been on the availability to crops of the N and P it contains and the soil conditioning capability of its organic matter content. The availability factor is the key to determining the fertilizer replacement value of sludge and thereby quantifying its agricultural benefit to farmers. Availability for agricultural crops, of the nitrogen from sewage sludge is in the range of 15–85%, compared to the availability of N from inorganic mineral fertilizers, and the availability of phosphorus for plants is about 50% in the majority of products from waste sludge. Thus, the agricultural benefit of sludge products has been defined as effectively as is possible for an organic material and many farmers use sludge products, recognizing their value and economic benefit [13,14].

As a result of the research carried out by numerous authors it has been demonstrated that these types of organic materials must be treated properly before being applied to the soil. Selection of sludge treatment process is concerned principally with factors such as stabilization, sanitization and volume control but it is also important, if the sludge is for agricultural use, to have a sludge product which farmers will want to apply to their land. These treatments are made with the purpose to ameliorate the biological transformation of the organic matter, for obtaining a stabilized organic material avoiding any potential risks for plants and soils when these products are used as organic amendments [15]. The composting process is a helpful method for producing a stabilized organic material which can in turn be used as a nutrient source and amendment for soil. This methodology is usually divided into two phases: active and maturation. The first is characterized by an intense microbial activity leading to the decomposition of the biodegradable material and to organic residues stabilization. The second (humification stage), is characterized by conversion of a part of the organic matter in humic substances [16]. Through both processes the final products are stabilized and the mature organic matter, rich in humic acids, represents the principal component and the stabile one of the organic matter that essentially contributes to soil fertility and health. Therefore, the amount, and in particular, the quality of the humic acids from compost are considered as important indicators of biological, chemical stability and maturity and constitutes a guarantee for safe application and performance of the compost into the soil. Adding the organic amendments (compost), through the

input of humic acids, exert major influences on the physico-chemical proprieties of the soil, for the purpose of modifying the relationship between acids and bases, with the effect of increasing the total cationic exchange capacity and buffering capacity of the soil, problems intensively studied in the last years [11,17,18].

Our paper aims at studying the possibility of using an innovative composite as fertilizer, realized by composting marine algae biomass together with sewage sludge and farmyard manure. For this purpose, a detail physico-chemical analysis has been performed on the composition of the three raw materials and also on the soil sampled at two depths. Similar analyses have been carried out on representative samples of soil treated with the compost as compared with untreated soil samples. The efficiency of applying this compost on the field at large scale was assessed by chromatographic determination of fatty acid content of the oleaginous plants cultivated.

## 2. Materials and Methods

The experimental design for the production of an innovative composite as a fertilizer, involved a series of logistical steps: (a) waste collection: of farmyard manure (FM) from some farms in Agigea, Constanţa County, sewage sludge (SS) from the Wastewater Treatment Plant from Eforie Sud, Constanţa County and marine algae (MA), collected from the Romanian shores of the Black Sea, between Agigea and South of Eforie; (b) the establishment of the experimental site for composting, on a waterproofed platform inside a farm; (c) the initial analysis of the nutrient and toxic content of the three wastes; (d) composting; (e) analysis of the agrochemical properties of compost variants; (f) selection of the site for field experiment (Agigea, Constanta county, Romania) and establishment of experimental variants, (g) monitoring of soil quality under the effect of applied amendments and the development of cultivated plants.

### 2.1. Primary Analyses on the Biosolid Wastes Used

In addition to farmyard manure widely used as soil amendment, the other two biosolid waste contain a range of nutrients necessary for the plants [19].

Thus, the three biosolid wastes have been assessed for the macro- and micronutrients (even heavy metals), total soluble salts and soluble ions contents (Tables 1–3), in order to substantiate the premise of their use to obtain a compost with benefits for soil fertility and plant development.

**Table 1.** Macro and micronutrients content of the biosolid waste used for composting.

| Material | TOC | TN | N-NO$_3$ | N-NH$_4$ | P$_{AL}$ | K$_{AL}$ | Fe | Mn | Cu | Zn | Cd | Pb |
|---|---|---|---|---|---|---|---|---|---|---|---|---|
| | % | | | | mg·kg$^{-1}$ | | | | | | | |
| SS | 24 | 2.4 | 13 | 297 | 622 | 872 | 31,618 | 391 | 124 | 541 | 1.3 | 54 |
| FM | 30 | 1.9 | 41 | 28 | 30 | 6540 | 9965 | 43 | 27 | 84 | 0.5 | 27 |
| | N | P | K | Ca | Mg | Na | Fe | Mn | Cu | Zn | Cd | Pb |
| MA | % | | | | | | mg·kg$^{-1}$ | | | | | |
| | 3.40 | 0.29 | 2.20 | 4.95 | 0.92 | 0.88 | 3913 | 385 | 11 | 39 | 0.45 | 9.3 |

**Table 2.** The total content of soluble salts and water-soluble ions.

| Material | TSC | HCO$_3^-$ | SO$_4^{2-}$ | Cl$^-$ | Ca$^{2+}$ | Mg$^{2+}$ | Na$^+$ | K$^+$ |
|---|---|---|---|---|---|---|---|---|
| | mg/100 g | | | | | | | |
| SS | 531 | 131 | 95 | 58 | 10 | 8 | 19 | 210 |
| FM | 658 | 177 | 356 | 22 | 36 | 21 | 32 | 15 |

Regarding the total soluble salt content, the farmyard manure and sewage sludge have moderate concentrations (531–658 mg/100 g), each of the ions being present, however, K$^+$ is detached from the farmyard manure and SO$_4^{2-}$ from sewage sludge (Table 2). In addition,

the percentage distribution of salts is equitable, highlighting bicarbonates in manure and sulfates in sewage sludge (Table 3).

**Table 3.** The percentage composition (%) of soluble salts.

| Material | $Ca(HCO_3)_2$ | $Mg(HCO_3)_2$ | $NaHCO_3$ | $KHCO_3$ | $CaSO_4$ | $MgSO_4$ | $Na_2SO_4$ | $K_2SO_4$ | $KCl$ |
|---|---|---|---|---|---|---|---|---|---|
| SS | 6.8 | 9.5 | 11.1 | 10.0 | - | - | - | 34.4 | 28.2 |
| FM | 26.5 | - | - | - | 7.6 | 32.3 | 26.6 | 1.3 | 5.7 |

*2.2. Composting Technology and Experimental Design of Application on Soil*

Four types of compost have been designed consisting of different ratios of the three biosolid wastes, namely: a variant with equal parts of three wastes, and other three variants that, in turn, contained 50% of each waste, to which was added 25% each of the other two wastes. Composting has been carried out into Könemann type cubes, with a side of 1.20 m, and 1.73 $m^3$ volume. Composting lasted two months, from 19 July to 16 September, both under reducing and oxidative conditions. In the reducing phase the temperature of the compost reached 63 °C, when the material has been sterilized. The temperature measurements were performed in three positions (the top of the mixture, the middle and bottom) three times a day (morning, midday and evening). In the oxidative phase, recorded temperatures were close to those of the ambiental environment. At the end of composting process, the obtained material was passed through a special mill, resulting in pellets.

The experimental field was located on a Cambic Chernozem in the northwestern part of Agigea (Constanța County, Romania) not far from the southern side of the Danube-Black Sea canal.

The soil in the experimental field was plowed to 30 cm and then homogenized with a disc harrow. In the spring, in March, the doses of amendments were incorporated, respectively, the 4 types of compost and mineral fertilizers, also with the help of the disc harrow, and two crops were sown, one of corn and another of sunflower. The PIONEER P 9241 hybrid, group 340 FAO, was sown for corn, and the P 64LE99 hybrid for sunflower, according to the manufacturers' instructions.

For analysis of composting effects on amended soils a number of 18 variants, each with three replicates, have been used, as presented in the Table 1, the first two being taken as control, unfertilized soil and soil chemically fertilized, while the other 16 samples consist in mixtures of soil with the four biomass variants in four equivalent doses: 25, 50, 75 and 100 t/ha (Table 4).

**Table 4.** Experimental treatments.

| Variant Number | Compozite Mixture | The Compost Dose Equivalent for the Field Application (t/ha) | Variant Number | Compozite Mixture | The Compost Dose Equivalent for the Field Application (t/ha) |
|---|---|---|---|---|---|
| V1 | | Control-unfertilized soil | | | |
| V2 | | Control-mineral N fertilized soil | | | |
| V3 | I st compost: 33.33% FM x), 33.33% SS x), 33.33% MA x) | 25 | V11 | III rd compost: 50% SS, 25% FM, 25% MA | 25 |
| V4 | | 50 | V12 | | 50 |
| V5 | | 75 | V13 | | 75 |
| V6 | | 100 | V14 | | 100 |
| V7 | II nd compost: 50% FM, 25% SS, 25% MA | 25 | V15 | IV th compost: 50% MA, 25% FM, 25% SS | 25 |
| V8 | | 50 | V16 | | 50 |
| V9 | | 75 | V17 | | 75 |
| V10 | | 100 | V18 | | 100 |

x) FM = farmyard manure, SS = sewage sludge, MA = marine algae.

### 2.3. Analytical Methods Used

Quantification of organic matter. Analytical methods applied were FT-IR spectroscopy in correlation with organic/inorganic and total carbon (TOC/TIC/TC) analysis. Physicochemical analysis has been completed by the analytical evaluation of organic matter and of humic acids. FT-IR spectra have been registered on a Perkin Elmer spectrometer in ATR technique. TC/TOC analysis has been performed on N/C Multianalyser, Analytic Jena.

Analysis of fatty acids content and their composition in maize grains. Weight amounts of maize grains have been continuously extracted for 6 h according to standard Soxhlet protocol [20], and the fatty acids content was calculated as percentage from initial mass. In order to assess the nutritive quality of the oil extracted, the fats (triglycerides) were converted in the corresponding fatty acids methyl esters (FAME) in order to apply gas chromatography method. FAME were prepared by transesterification of oils with methanol, using $BF_3$-MeOH complex as catalyst, according to the standard method [21]. This catalyst complex allows complete fatty acids esterification, from both triacyglycerols and free fatty acids. The chromatography standard SupelcoTM37 Component FAME Mix was purchased from Supelco, petroleum ether 40–60 °C fraction (analytical purity grade) and $CH_2Cl_2$ (HPLC purity grade) were purchased from Sigma-Aldrich, methanol (HPLC isocratic grade) and $BF_3$-MeOH were purchased from J.T. Baker and Alpha Aesar, respectively. The gas-chromatograms of FAME were recorded in triplicate on an Agilent Technologies 7890A instrument equipped with Agilent auto sampler and Triple Axis mass detector model 5975C VL MSD, under the following operating conditions: oven temperature 140 °C for 5 min, then 4 °C/min to 240 °C were maintained for 20 min (a total routine of 50 min), carrier gas He, split ratio 100: 1. Separation into components was carried out on a capillary column dedicated for FAME analysis (SupelcoTM 2560: 100 m length, 0.25 mm inner diameter, 0.2 μm film thickness).

## 3. Results

### 3.1. Physical Chemical Characterization of Soil and Composting Variants Used

After the completion of the composting process, analyzes were performed to establish the main agrochemical parameters of the four compost variants obtained, compared to those of the soil from the future experimental field (Tables 5–7).

**Table 5.** The main agrochemical properties of experimental soil and the compost in the final phase.

| Material | pH | TOC. | TN | C/N | N-NO$_3$ | N-NH$_4$ | P$_{AL}$ * | K$_{AL}$ |
|---|---|---|---|---|---|---|---|---|
| | H$_2$O | % | | | mg·kg$^{-1}$ | | | |
| 1MA + 1SS + 1FM | 8.09 | 11.92 | 0.99 | 14.0 | 439 | 89 | 181 | 5983 |
| 2FM + 1MA + 1SS | 8.10 | 12.06 | 0.88 | 15.9 | 1224 | 76 | 187 | 5833 |
| 2SS + 1MA + 1FM | 7.92 | 13.78 | 1.09 | 14.6 | 2243 | 105 | 196 | 5517 |
| 2MA + 1SS + 1FM | 7.82 | 11.09 | 0.94 | 13.8 | 1124 | 74 | 205 | 4383 |
| Soil | 7.52 | 3.25 | 0.25 | 15.0 | 143 | 6 | 266 | 700 |
| Mean $_{Compost}$ | 7.98 | 12.21 | 0.98 | 14.58 | 1257.50 | 86.00 | 192.25 | 5429.00 |
| STDEV | 0.14 | 1.13 | 0.09 | 0.95 | 743.89 | 14.31 | 10.50 | 723.88 |
| Compost/soil | 1.06 | 3.76 | 3.90 | 0.97 | 8.79 | 14.33 | 0.72 | 7.76 |

* values recalculated according to the reaction of the material.

Chemical analyses performed in the final stage of composting, revealed a significant decrease in the organic carbon content, and a slight increase in total nitrogen content, that led to C/N value of 14.6, close to a normal soil. It was also noted a sharp increase in nitric nitrogen content and a decrease of ammonia nitrogen. They have individualized very high values of mobile forms of P and K, and the normal values of trace elements and heavy metals contents, with a tendency to increasing the micronutrients content and lowering the Cd and Pb content. It is also noted a high content of soluble salts, consisting essentially of magnesium and sodium sulfates, and potassium chloride. However, higher levels of

soluble salts in the compost in the final phase will not have a significant influence on the mixture of soil and compost, because it will interfere with a high dilution coefficient of soluble salts in the soil [19].

**Table 6.** The content in microelements and heavy metals (mg kg$^{-1}$) of experimental soil and the compost in the final phase.

| Material | Cd | Co | Cr | Cu | Fe % | Mn | Ni | Pb | Zn |
|---|---|---|---|---|---|---|---|---|---|
| 1MA + 1SS + 1FM | 0.54 | 6.74 | 102 | 43 | 1.66 | 462 | 38 | 53 | 200 |
| 2FM + 1MA + 1SS | 0.47 | 6.66 | 102 | 37 | 1.71 | 508 | 38 | 63 | 151 |
| 2SS + 1MA + 1FM | 0.55 | 7.73 | 82 | 40 | 1.61 | 472 | 33 | 46 | 188 |
| 2MA + 1SS + 1FM | 0.46 | 5.51 | 81 | 42 | 1.50 | 410 | 30 | 50 | 166 |
| Soil | 0.23 | 12.8 | 39 | 29 | 2.90 | 816 | 38 | 29 | 115 |
| Mean $_{Compost}$ | 7.98 | 12.21 | 0.98 | 14.58 | 1257.50 | 86.00 | 192.25 | 5429.00 | 7.98 |
| STDEV | 0.14 | 1.13 | 0.09 | 0.95 | 743.89 | 14.31 | 10.50 | 723.88 | 0.14 |
| Compost/soil | 2.20 | 0.52 | 2.35 | 1.40 | 0.56 | 0.57 | 0.91 | 1.83 | 1.53 |

**Table 7.** Total soluble salt and water-soluble ion content of experimental soil and the compost in the final phase.

| Material | TSS | HCO$_3^-$ | SO$_4^{2-}$ | Cl$^-$ | Ca$^{2+}$ | Mg$^{2+}$ | Na$^+$ | K$^+$ |
|---|---|---|---|---|---|---|---|---|
| | | | | mg/100 g material | | | | |
| 1MA + 1SS + 1FM | 2358 | 102 | 981 | 396 | 102 | 117 | 250 | 410 |
| 2FM + 1MA + 1SS | 2216 | 119 | 886 | 372 | 92 | 92 | 228 | 428 |
| 2SS + 1MA + 1FM | 2891 | 195 | 1594 | 614 | 126 | 187 | 418 | 429 |
| 2MA + 1SS + 1FM | 2500 | 105 | 1164 | 392 | 124 | 135 | 280 | 301 |
| Soil | 42 | | | | | | | |
| Mean $_{Compost}$ | 2491 | 130 | 1156 | 444 | 111 | 133 | 294 | 392 |
| STDEV | 290.6 | 43.8 | 313.8 | 114.2 | 16.7 | 40.2 | 85.4 | 61.3 |
| Compost/soil | 59.3 | | | | | | | |

Values recalculated according to the reaction of the material.

It is important to note that analyzes of organic carbon fractions in the four compost variants showed that 98–99% of organic C comes from humines, which indicates the high quality of composted material as a potential reservoir of nutrients for plants (Table 8).

**Table 8.** The organic carbon fractions in the experimental soil and the compost in the final phase.

| Material | TOC | C from Humines | C from Humic Acids | C from Fulvic Acids |
|---|---|---|---|---|
| 1MA + 1SS + 1FM | 12.8 | 12.7 | 0.022 | 0.079 |
| 2FM + 1MA + 1SS | 12.9 | 12.8 | 0.034 | 0.077 |
| 2SS + 1MA + 1FM | 14.8 | 14.6 | 0.030 | 0.089 |
| 2MA + 1SS + 1FM | 11.9 | 11.8 | 0.022 | 0.075 |
| Soil | 3.5 | 3.41 | 0.012 | 0.056 |
| Mean $_{Compost}$ | 13.1 | 13.0 | 0.027 | 0.080 |
| STDEV | 1.2 | 1.2 | 0.006 | 0.006 |
| Compost/soil | 3.7 | 3.8 | 2.3 | 1.4 |

The analytical data of the soil selected for the field experiment is a Cambic Chernozem, located in Agigea locality, Constanţa county (near Black Sea shoreline), slightly alkaline

(pH 7.52), with medium humus content (5.60%), medium-low total nitrogen (0.25%), but high content of nitric nitrogen (N-NO$_3$) and a normal C/N ratio.

In addition, the soil has high contents in mobile forms of P and K, soluble in ammonium acetate-lactate at pH 3.7 and total content of trace elements and heavy metals with normal values, except Co and Zn with values slightly higher than normal and very low soluble salt content (42 mg/100 g soil).

Alluvial soil is to be cultivated with corn (*Zea mays*) and sunflower (*Helianthus annus*) plants.

### 3.2. Analysis of Organic Matter in Soil, Biomasses and Composting Variants

In Figure 1 the IR spectra are represented for two soil samples from the experimental field taken from two depths: sample 1 between 20–40 cm and sample 2 between 0–20 cm, which are compared with the sample 3 for humic acid as reference compound.

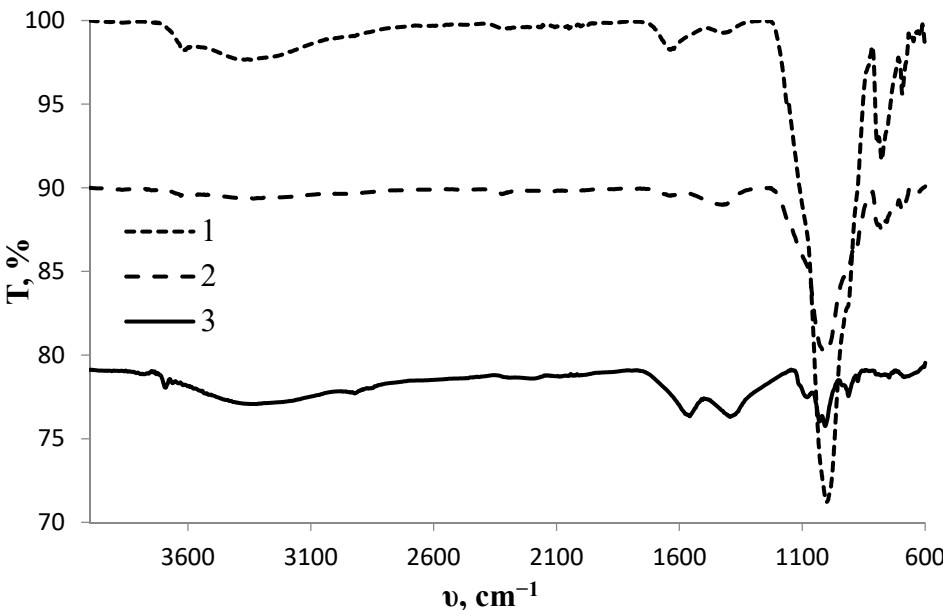

**Figure 1.** IR spectra: 1. 20–40 cm depth soil; 2. 0–20 depth soil; 3. humic acid.

The most intensive band corresponds to the mineral component of alumino-silicate origin, evidenced by specific Si-O vibration at 1000 cm$^{-1}$, which overlaps the bands of oxidic groups from humic acids. However, the presence of organic matter existent in soil samples is confirmed by the large band from the region 3300–3400 cm$^{-1}$, coupled with the weak bands at 1638 cm$^{-1}$ of carbonyl groups (C = O) and 1420 cm$^{-1}$ of double bonds (C = C), which are better evidenced for the depth sample than for surface sample. This qualitative assessment is confirmed by carbon analyses (TC, TOC, TIC) collected in the Table 9.

Indeed, the two soil samples have comparable total carbon content (TC = 18.50 and 20.62, respectively), but while for the surface soil (sample 2) the inorganic carbon (TIC = 15.58) is dominant over the organic carbon (TOC = 5.04) as expected, for the depth sample (sample 1) TOC ≥ TIC. This result is important as usually the plant roots are at 20–40 cm depth. The organic source of the soil is mainly provided by the humic acids as can be seen from Figure 1, number 3. The characteristic bands at 1556 cm$^{-1}$ for double bonds and 1395 cm$^{-1}$ for organic oxidic compounds from humic acids are shifted towards higher wave numbers, but more evident in the depth sample than in the surface soil sample.

TOC analysis was also useful to decide which sewage sludge to be chosen from two treatment plant locations: Constanta or Eforie. By comparing these two samples (SS-Ct and SS-Ef from Table 9) one may conclude that the second source is more convenient due to a higher organic carbon content and a lower mineral contribution. This observation is also strongly supported by IR analysis, as it is obviously from the Figure 2, where the IR spectra for the three biomasses have been represented taking the unfertilized soil as Control 1, and soil chemically treated with ammonium nitrate as Control 2. The presence of nitrate anion

in this last sample is demonstrated by the most intensive band of the IR spectra from the Figure 2, at 1036 cm$^{-1}$ provided by $\upsilon_1$ vibration mode.

**Table 9.** TC/TOC/TIC analyses.

| Sample | TC, g/kg | TOC, g/kg | TIC, g/kg | TOC/TIC |
|--------|----------|-----------|-----------|---------|
| Soil 1 | 18.50 | 9.51 | 8.99 | 1.06 |
| Soil 2 | 20.62 | 5.04 | 15.58 | 0.32 |
| FM | 304.10 | 244.90 | 59.20 | 4.13 |
| SS-Ct | 232.33 | 179.50 | 52.83 | 3.40 |
| SS-Ef | 246.60 | 198.00 | 48.60 | 4.07 |
| MA | 546.80 | 521.40 | 25.40 | 20.53 |
| C-I | 23.25 | 12.28 | 10.97 | 1.12 |
| C-II | 25.88 | 19.46 | 6.42 | 3.03 |
| C-III | 27.16 | 11.65 | 15.51 | 0.75 |
| C-IV | 28.43 | 22.11 | 6.32 | 3.49 |

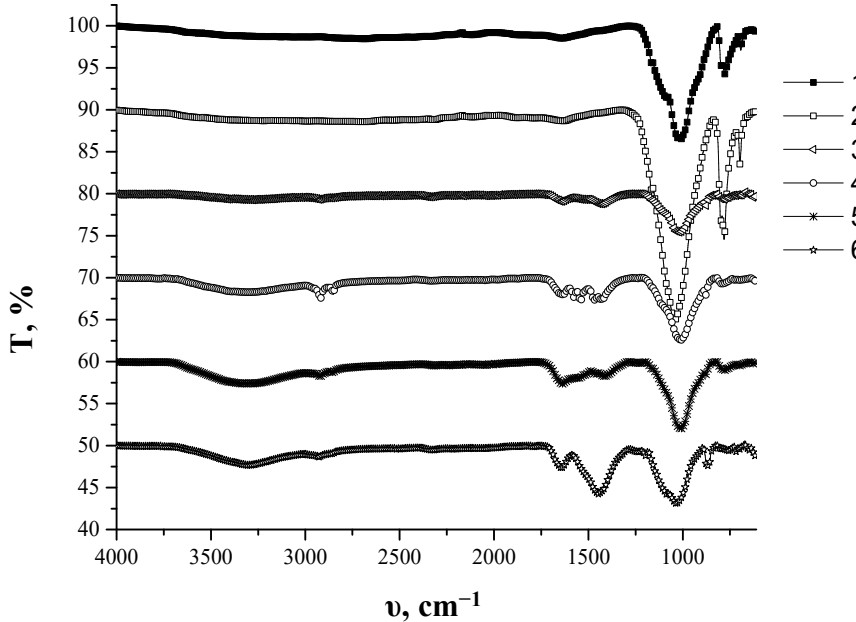

**Figure 2.** IR spectra: 1. Control 1-unfertilized soil; 2. Control 2-fertilized soil with mineralized N; 3. Constanta sludge; 4. Eforie sludge; 5. Farmyard manure; 6. Marine algae.

The two sludge samples (Figure 2, samples 3 and 4) show similar vibration bands, by means of the most intensive band at 1008 cm$^{-1}$ corresponding to C–O groups from phenols, carboxylic acids, ester structures, mostly overlapped by very intensive Si–O band from alumino-silica minerals, accompanied by the double band at 778–791 cm$^{-1}$ specific to quartz. It is also present the large band at 3200–3400 cm$^{-1}$ corresponding to vibrations at O–H and N–H groups. However, the two sludge samples exhibit some specific bands which differentiate them: in the Eforie sludge it is very clear the doublet at 2916–2847 cm$^{-1}$ as fingerprint for methylene groups, accompanied by a five close bands in the region 1400–1600 cm$^{-1}$ assessed to most organic compounds containing carbonyl, double bonds, amines, amides, etc. One may conclude that Eforie sludge is richer in organic components than Constanta sludge, as it was also confirmed by TC, TOC, TIC analyses mentioned above. For this reason, the sewage sludge from Eforie has been used in further experiments.

As referring to the last two biomasses used (samples 5 and 6 in Figure 2) for farmyard manure and marine algae, respectively, they seem to have similar patterns. However, two distinct regions make them identifiable in compost mixtures. One of them is around 1000 cm$^{-1}$ where there are many oxygen containing groups, which is larger in algae than in manure samples, and the pair of bands in the region 1400–1650 cm$^{-1}$. Thus, for algae the band at 1444 cm$^{-1}$ is dominant due to the presence of calcite from shells ($\upsilon_3$ vibration mode) as compared to the band at 1636 cm$^{-1}$ assessed to carbonyl and double bond groups. On the contrary, in the IR spectrum of manure the band at 1635 cm$^{-1}$ is dominant due to complementary contribution of Amide I band from proteins, as compared to the band at 1542 cm$^{-1}$, where also Amide II band is present.

Analysis of compost variants. For preparing variants of composts, different mixtures of soil with the three biomasses in proportions mentioned in the Table 1 have been obtained and analyzed.

In Figures 3–5 the soil samples treated with the variants of compost where one of biomasses is in dominant proportion (50%) applied in the maximum dose (100 tone/ha) with the corresponding dominant biomass are compared. Thus, in Figure 3 the variant V10 is compared with farmyard manure, showing similar spectral patterns, with dominance of the band at 1635 cm$^{-1}$ over its pair at 1542 cm$^{-1}$.

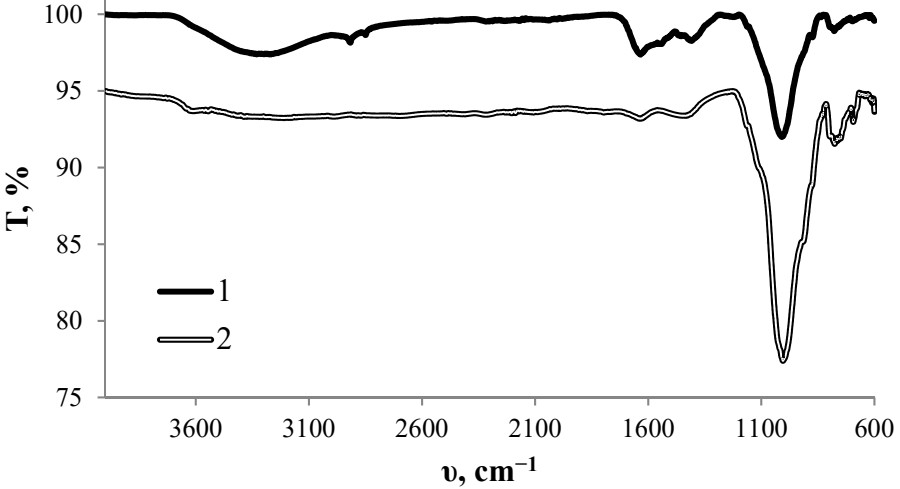

**Figure 3.** IR spectra: 1. Farmyard manure; 2. V10.

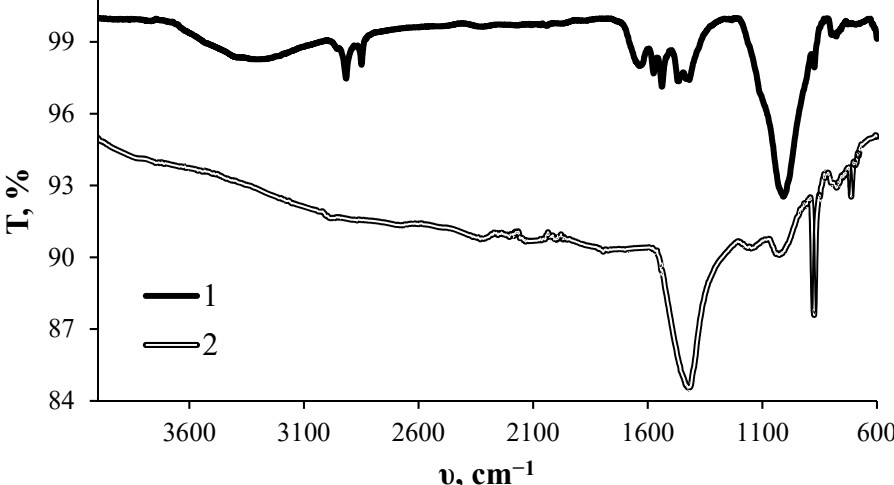

**Figure 4.** IR spectra: 1. Eforie sludge; 2.V14.

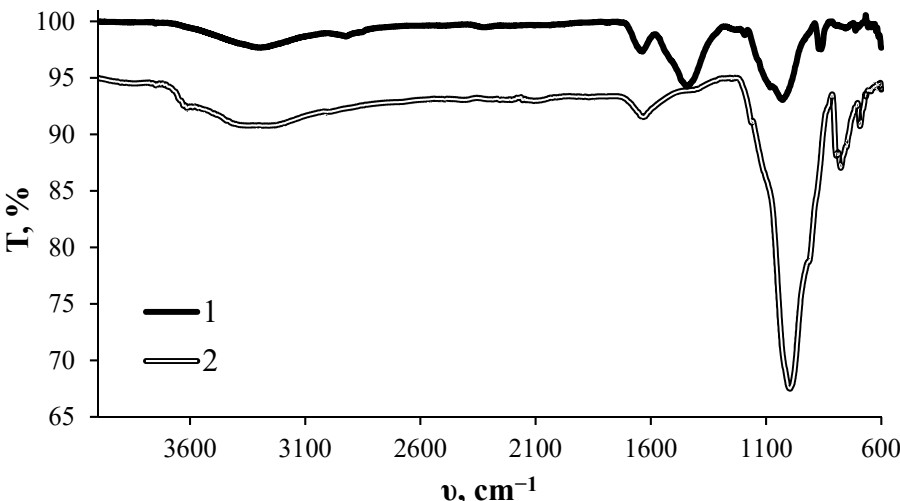

**Figure 5.** IR spectra: 1. Algae; 2. V18.

A similar analysis is made in Figure 4 between the sewage sludge and variant V14 at maximum dose of sludge in compost, but in this case the two spectra are quite different, as the complex structure of bands specific to amidic compounds from sludge is replaced by a strong band centered at 1420 cm$^{-1}$ indicating protein degradation during composting process and formation of oxygenated groups.

Some noticeable changes appear also when compare algae biomass with the variant V18, as the dominant band of calcite at 1444 cm$^{-1}$ from algae becomes smaller in the compost, where the band at 1631 is more intensive, while the most intensive band at 998 cm$^{-1}$ becomes narrow as in sludge.

Carbon analyses of these compost variants presented in Table 1, reveals a superior carbon content as compared with control samples. At the same time the balance between organic and inorganic carbon shows a significant improvement in the variants containing farmyard manure and algae, while for sludge their ratio is a bit in favor of mineral components.

## 4. Discussion

The soil structure should be suitable for the germination of the seeds and the growth of the roots and must have characteristics that enhance the storage and supply of water, nutrients, gases and heat to the crop. Soil chemistry is dominated by the interaction between its solid components (primarily the insoluble compounds of silica, calcium and aluminum) and its water phase. Understanding soil chemistry is of paramount importance, since it is the basis of soil fertility and provides the needed knowledge to understand the differences in fertility among different soils and their response to fertilization. Sometimes soil chemistry can have a direct impact on soil physical conditions as in the case of basic soils with high exchangeable sodium content [22].

Fertilization practices usually used in agricultural systems include the application of chemical nitrogen (N), phosphorus (P) and potassium (K) fertilizers, organic fertilizers and animal manures or straw, or combinations of chemical fertilizers and organic sources of nutrients. N, P, and K are macronutrients necessary for crop growth and the application of the chemical fertilizers can affect soil C and N accumulation through increasing biomass production [23].

Organic carbon concentrations in soil depend on the amounts of fertilizers applied to soil, their chemical composition, and decomposition rate [24]. Application of organic fertilizers or combinations of chemical and organic fertilizers can further increase soil organic carbon (SOC) and total nitrogen (TN) concentrations because of enrichment with C sources and additional macronutrients and micronutrients present in organic fertilizers and especially in manures [15]. Overall, the storage of SOC or TN depends on the input

and output balance of C or N when other nutrients are present at concentrations that do not limit crop growth.

In our fertilization strategy followed for two years we proposed an innovative biocomposite by composting mixtures of three biosolid wastes with significant organic matter content, that have been applied on agricultural lands in four doses: 25, 50, 75, and 100 tones/ha. The efficiency of these variants applied for maize and sunflower cultures has been assessed by means of the oil content and of their composition in fatty acids obtained from the crops harvested from amended soils as compared to not amended soil and the amendment with chemical NPK fertilizer, as control samples.

### 4.1. Effect of Compost Application on Field Cultures

The results obtained for oil content in maize cultivated on three lots are collected in the Table 10. One may observe that in all the soils amended with compost, the oil content is superior as compared with the reference soils, with slight dependence on the dose applied. Moreover, the values of the oil content are higher in the second year *versus* the first year, as effect of successive fertilization. When compare the results obtained for the three compost variants, the highest efficiency is obtained for CIV variant with higher proportion of algae, The preference of maize culture for this compost variant may be explained by the balanced contribution in both organic and inorganic nutrients provided by this valuable biomass.

**Table 10.** Oil Content in Maize Grains Harvested in Two Successive Years *.

| Compost Variant | Dose t/ha | Oil Content, % I $^{st}$ Year | Oil Content, % II $^{nd}$ Year |
|---|---|---|---|
| - | | 2.25 | 2.30 |
| NPK | | 2.04 | 2.14 |
| CII | 25 | 2.37 | 2.82 |
| CII | 50 | 2.46 | 2.80 |
| CII | 75 | 2.57 | 2.92 |
| CII | 100 | 2.60 | 2.92 |
| CIII | 25 | 2.59 | 2.89 |
| CIII | 50 | 2.85 | 2.90 |
| CIII | 75 | 2.89 | 2.92 |
| CIII | 100 | 2.90 | 2.90 |
| CIV | 25 | 2.73 | 3.74 |
| CIV | 50 | 2.90 | 3.79 |
| CIV | 75 | 3.05 | 3.81 |
| CIV | 100 | 3.20 | 3.61 |

* Values are averages for three lots, standard deviation, $\sigma = \pm 0.05\%$.

Similar experiments conducted on sunflower cultures led to higher oil content as expected, ranging between 43–48%, but the best efficiency was obtained for the lots amended with the compost containing mostly farmyard manure, the organic matter contribution being more significant in this case.

### 4.2. Nutritive Quality of the Oil Composition of Maize Grains

In order to assess nutritive value of these crops, a detail GC-MS analysis was carried out on the fatty acids composition of the oil extracted from maize grains. The results showed in Table 11 reveal a high content of unsaturated fatty acids (87–88%), and the dominant presence of linoleic acid ($\omega$6), with a very high nutritive value for human health.

**Table 11.** Fatty Acids Composition of Maize Crop of the Second Year, %.

| Compost Variant/ Dose | Palmitic | Stearic | Oleic ω9 | Linoleic ω6 | Linolenic ω3 | Total Unsaturated Fatty Acids % |
|---|---|---|---|---|---|---|
| Not amended | 11.05 | 1.49 | 30.50 | 56.24 | 0.72 | 87.46 |
| NPK | 11.04 | 1.47 | 30.31 | 56.14 | 0.84 | 87.39 |
| CII/25 | 11.09 | 1.53 | 29.42 | 57.10 | 0.85 | 87.38 |
| CII/50 | 10.90 | 1.61 | 30.56 | 55.97 | 0.95 | 87.49 |
| CII/75 | 10.92 | 1.41 | 20.31 | 57.66 | 0.69 | 87.57 |
| CII/100 | 10.73 | 1.33 | 30.53 | 56.94 | 0.49 | 87.94 |
| CIII/25 | 10.47 | 1.43 | 30.22 | 57.12 | 0.75 | 88.10 |
| CIII/50 | 10.46 | 1.31 | 30.11 | 57.89 | 0.22 | 88.11 |
| CIII/75 | 11.28 | 1.54 | 29.65 | 57.64 | 0.90 | 88.18 |
| CIII/100 | 10.17 | 1.22 | 30.13 | 57.45 | 0.13 | 88.61 |
| CIV/25 | 11.04 | 1.43 | 30.57 | 56.20 | 0.76 | 87.57 |
| CIV/50 | 10.86 | 1.50 | 29.77 | 57.12 | 0.75 | 87.64 |
| CIV/75 | 10.76 | 1.51 | 30.82 | 56.06 | 0.86 | 87.73 |
| CIV/100 | 10.83 | 1.40 | 30.69 | 56.32 | 0.76 | 87.76 |

Surprisingly, the highest proportion on total unsaturated fatty acids content is obtained for the soils amended with the compost variant containing mostly sewage sludge, thus confirming also its balanced proportion between organic and inorganic nutrients, so important for maize culture. This observation is of high economic and ecological importance, as it offers solutions to use huge amounts of sewage sludge waste resulted from water treatment plants for optimal soil remediation.

## 5. Conclusions

An innovative polycomposite is proposed as fertilizer for soil remediation by using three organic wastes (manure, sewage sludge and seaweed), in a mixture of equal parts of each waste and in mixtures containing, in turn, 50% of one of the wastes and the other in proportions of 25% each, four composting variants were obtained depending on the quantitative ratio between the three components.

The analyzes performed revealed that this waste contains macroelements in large quantities, microelements (Cu, Zn) in slightly increased quantities, and heavy metals in small-normal quantities.

During composting, thermal values of up to 63 °C were reached, which contributed to the sterilization of the material, respectively, the removal of potentially pathogenic microorganisms from manure or sewage sludge.

Composting, carried out in Könemann type cubes, with a side of 1.20 m, and 1.73 $m^3$ volumes consisted of alternating the oxidative phase with the reducing one, over a period of approximately 60 days, during which the temperature and humidity of the mixtures were monitored, daily.

In order to follow structural changes during composting process, a systematic study was performed on the raw materials, as well as final analyzes of the composted mixtures to assess for the macro- and micronutrients (even heavy metals), total soluble salts and soluble ions contents.

The total organic carbon fractions in the four compost variants showed that 98–99% of organic C comes from humines, which indicates the high quality of composted material also as a potential reservoir of nutrients for plants and an excellent ecologic service provider to soil carbon sequestration.

For analysis of organic/inorganic matter balance two investigation methods have been used: FT-IR and TC/TOC/TIC analyses. Corroboration of the results obtained by these two methods allowed assessing relative proportion between organic and mineral components. Fingerprints have been identified to recognize the presence of various biomasses in the variants of compost, and possibly to follow their evolution during 3–4 years of plants growing cycles.

Finally, four variants of compost of good nutritional quality for plants were obtained whose effects on soil fertility and stimulating the development of cultivated plants was tested in the experimental field, on a Cambic Chernozem soil, in experimental variants represented by the application of the four types of compost in doses of 25, 50, 75 and 100 t/ha and two control variants, one of which is mineral fertilized with N150 kg active substance/ha.

The efficiency of compost application on fields cultivated with corn and sunflower has been demonstrated by determination of oil content extracted from the seeds. The experiments on field revealed that the highest oil contents are obtained when the soil is amended with composts with major proportion of algae in case of corn and with farmyard manure for sunflower. Moreover, the best nutritive quality of the unsaturated fatty acids extracted from maize grains, expressed by maximum rate of linoleic acid ($\omega$6), was obtained in case of the compost with sewage sludge as major component.

One may conclude that the new bio-composite fertilizer can be successfully used for poor soils amendment cultivated with corn, when compost contains algae biomass or sewage sludge as major constituents, while they are applied above the minimum dose tested of 25 t/ha.

**Author Contributions:** Conceptualization, A.R.L., A.M. and C.C.; methodology, A.R.L., A.M. and C.C.; formal analysis, A.R.L., A.M., A.E.R. and C.C.; investigation, A.R.L. and C.C.; resources, A.R.L.; data curation, C.C.; formal analysis, A.R.L., A.M., A.E.R. and C.C.; writing—original draft preparation, A.R.L., A.M. and C.C.; writing—review and editing, A.R.L., A.M., A.E.R. and C.C.; supervising, A.R.L. and A.M., project administration, A.R.L., funding acquisition, A.R.L. All authors have read and agreed to the published version of the manuscript.

**Funding:** The received funding from Ministry of Research and Innovation- Executive Agency for Higher Education, Research, Development and Innovation (UEFISCDI), project no. PNII-PT-PCCA-2013-4-0675—FEROW and project no. PN-III-P1.2-PCCDI-2017-0721—INTER-ASPA.

**Institutional Review Board Statement:** Not applicable.

**Informed Consent Statement:** Not applicable.

**Data Availability Statement:** Exclude.

**Conflicts of Interest:** The authors declare no conflict of interest.

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
