# Peer review of "Innovative Polycomposite Fertilizer Obtained by Recycling and Processing Three Organic Wastes"

_agriculture, doi:10.3390/agriculture11101021_

Round 1
Reviewer 1 Report
The article entitled "Innovative polycompozite fertilizer obtained by recycling and processing three organic wastes" (agriculture-1387122) need major revisions.
The manuscript is well written and the authors justify clearly the study in the introduction by a good literature review. The section of "materials and methods" must be improved by reporting the main chemical characteristics of the three organic wastes and the respective compost. Moreover, it would suitable to report how the biomasses are mixed to soil. The experimental design must be better described, as well as the number of samples analysed and the sampling time.
The statistical anlyses section is missed.
The discussion of results and adeguate bibliography are completely missed. The importance of studying the different compost variatons is not noticeable as also the implications after their agricultural reuse as fertilizers.
Author Response
Dear Reviewer 1
Thank you for valuable recomendations and objective observations. We have completed all the sections with numerous experimental data and adequate comments in order to improve scientific quality of the paper and thus to correspond to publication standards. All the material completed is inserted in red in the original manuscript, and the answers to reviewer comments are briefly presented below.
Carolina Constantin

Reviewer 2 Report
The article entitled “Innovative polycomposite fertilizer obtained by recycling and processing three organic wastes " presents the results of research aimed at assessing the effect of marine algae compost, sewage sludge and farmyard manure on the content of carbon forms in soils and biomass.
General note
The use of various organic wastes for fertilization purposes has recently been quite popular. Much research is conducted in this area. Of course, each waste has a different chemical composition and therefore a different fertilizing value. For this reason, this type of research is needed. Unfortunately, in the case of the peer-reviewed article, there is no description of the conditions during the composting process, and in addition, the authors presented very few parameters tested in the compost and in the compost-treated soil. This significantly reduces the value of the article.
Throughout the manuscript, the authors incorrectly refer to the literature. This is against the guidelines for the authors. The same goes for the References part. Authors must correct this according to the format adopted in the journal.
All figures must be corrected. They are of poor quality.
Specific comments
Line 9: There is: “an innovative ecological organic fertilizer” – in my opinion “ecological” must be delated. A fertilizer from sewage sludge, cannot be called ecological.
At this point, I want to emphasize that “farmyard manure “ it is not organic waste, but organic fertilizer!
Line 10 – there is: “residual marine biomass”, while the rest of the article talks about the biomass of marine algae. In abstract, it is better to emphasize that algae were investigated and not marine residues.
Lines 110 - 112 - Wrong font size
There is: “The efficiency of applying this compost on the field at large scale was assessed by determination of fatty acid content of the oleaginous plants cultivated” - The methodology does not describe this growing experiment. It is only stated that the soil was mixed with the compost, but where, when the experiment was set up, how many plants were sown, how long were cultivated?
No description of the methodology for determining fatty acid content.
how many repetitions was the experiment conducted?
Line 104 – 106 – please rephrase
MATERIALS AND METHODS
This part needs to be completed taking into account the remarks above
No statistical analysis of the results
RESULTS AND DISCUSSION
Table 2 - Please delete column 1
Despite the name: "RESULTS AND DISCUSSION" - this part does not contain discussions with the literature at all – please add discussion part.
Where are the results of these measurements "was assessed by determination fatty acid content of the oleaginous plants cultivated"!!??
CONCLUSION
Lines 224 – 233 these are not conclusions but a summary. It has to be deleted and part of the conclusions has to be rewritten.
Lines 234 – 236 - it does not result from the conducted research at all, so you cannot write about it in the conclusions
Author Response
Dear Reviewer 2,
The authors express respectful thanks for careful and detail analysis of our manuscript and for useful advice given for substantial improvement of the paper, in order to meet journal requirements for publication.
The entire text has been properly revised and requested completions on experimental data and adequate comments have been inserted in red in the original manuscript.
A brief response to reviewer answers is given below in order to indicate the revised parts into initial manuscript.
Carolina Constantin

Round 2
Reviewer 2 Report
I appreciate the authors' efforts on this manuscript, which indeed improve the quality of this manuscript. Particularly, the authors added missing information in Materials and Methods, updated data. Thus, I satisfy the authors' respondence and the revision.